# SinglePointRNA, an user-friendly application implementing single cell RNA-seq analysis software

Laura Puente-Santamaría[1,2¤]*, Luis del Peso[1,3,4,5]

1 Departamento de Bioquímica, Facultad de Medicina, Universidad Autónoma de Madrid (UAM), Madrid, Spain, 2 Genomics Unit Cantoblanco, Fundación Parque Científico de Madrid, Madrid, Spain, 3 IdiPaz, Instituto de Investigación Sanitaria del Hospital Universitario La Paz, Madrid, Spain, 4 Centro de Investigación Biomédica en Red de Enfermedades Respiratorias (CIBERES), Instituto de Salud Carlos III, Madrid, Spain, 5 Unidad Asociada de Biomedicina CSIC-UCLM, Albacete, Spain

¤ Current address: Biocomputing Unit, Instituto Aragonés de Ciencias de la Salud, Zaragoza, Spain
* laura.puentes@estudiante.uam.es

**Data Availability Statement:** The software described in this manuscript, as well as all datasets used in the accompanying documentation, are available for download through GitHub at the

## Abstract

Single-cell transcriptomics techniques, such as scRNA-seq, attempt to characterize gene expression profiles in each cell of a heterogeneous sample individually. Due to growing amounts of data generated and the increasing complexity of the computational protocols needed to process the resulting datasets, the demand for dedicated training in mathematical and programming skills may preclude the use of these powerful techniques by many teams. In order to help close that gap between wet-lab and dry-lab capabilities we have developed SinglePointRNA, a shiny-based R application that provides a graphic interface for different publicly available tools to analyze single cell RNA-seq data. The aim of SinglePointRNA is to provide an accessible and transparent tool set to researchers that allows them to perform detailed and custom analysis of their data autonomously. SinglePointRNA is structured in a context-driven framework that prioritizes providing the user with solid qualitative guidance at each step of the analysis process and interpretation of the results. Additionally, the rich user guides accompanying the software are intended to serve as a point of entry for users to learn more about computational techniques applied to single cell data analysis. The SinglePointRNA app, as well as case datasets for the different tutorials are available at www.github.com/ScienceParkMadrid/SinglePointRNA.

## Introduction

The lowering prices and wider availability of massive sequencing and single cell analysis techniques have increased the volume and complexity of data generated in life sciences research. In this context interdisciplinary collaboration becomes essential for projects to yield results, making communication and coordination between researchers of very different backgrounds a fundamental necessity for knowledge generation in life sciences. Single-cell RNA-seq analysis requires both training in statistics and computation, as well as detailed knowledge on the

following repository: www.github.com/ScienceParkMadrid/SinglePointRNA Within the aforementioned repository, the folder "SinglePointRNA" contains the code and data needed to run the app. The folder "Tutorial_Datasets" contains the datasets for the three case studies depicted in the "User's Guide" section of the app.

**Funding:** This research was funded by Ministerio Ciencia e Innovacion (MCIN/AEI/10.13039/501100011033 "ERDF A way of making Europe", Spain) grant number PID2020-118821RB-I00, awarded to Luis del Peso, and Consejeria de Ciencia, Universidades e Innovacion de la CAM (Madrid, Spain) grant number IND2019/BMD-17134, awarded to Luis del Peso and Laura Puente-Santamaría. The funders had no role in study design, data collection and analysis, decision to publish, or preparation of the manuscript.

**Competing interests:** The authors have declared that no competing interests exist.

biological background of the dataset examined. As a consequence, there is an increasing need to integrate mathematically and computationally demanding processes into what, until recently, were mainly wet-lab based studies. This sets out the need to create bridges between disciplines that for a long time have been taught detached from each other throughout most educational stages. Beyond interdisciplinary collaboration, another pathway to further assist in this process is to make the computational tools more accessible to experimental researchers.

The goal of the work described here is to develop a tool that can provide reliable options for knowledge discovery to researchers in life sciences without the need for extensive preparation in computation, programming, or machine learning. Given that many users in the target audience might have been dissuaded from delving deeper into bioinformatics or computer science in general, in addition to give researchers autonomy in processing the data they've generated, this application aims to work as a point of introduction to data science and machine learning techniques involved in scRNA-seq processing.

To this end the tool provides a menu-driven interface providing access to widely used scRNA-seq analysis tools such as the Seurat R package [3] and scCATCH [4]. SinglePointRNA implements the basic steps needed for single-cell RNA-seq analysis after read mapping and transcript quantification: quality measurement and filtering of low-quality cells, cell cycle phase estimation and regression, dimensionality reduction and cell clustering, cell type imputation, differential expression, and pathways analysis. In addition, we developed additional functions not included in these packages. Specifically, SinglePointRNA includes a graphical analysis that guide the selection of clustering parameters, a key and non-trivial step in the analysis. We also included functions that reflect the uncertainty of cluster membership which, indirectly, informs about the reliability of the clusterization and hence the suitability of the chosen parameters. Finally, we included functions to perform singular enrichment analysis on the list of genes defining a cluster to provide biological insight from the analysis' results.

Additionally, the application is designed to guide the user throughout the analysis and includes rich documentation, rather than a bare bone description of the parameters, explaining the procedures performed at each step of the process, as well as an introduction to their theoretical basis.

SinglePointRNA strives to find a balance between providing an automatically optimized pipeline and giving freedom for users to alter analysis parameters once they feel more comfortable in their knowledge of the subject. This application also includes comprehensive guidance in every step of the process, from help messages in most input widgets to comprehensive tutorials, in addition to introductory lessons on machine learning techniques tailored to users from a mainly experimental life sciences background.

## Materials and methods

### Software dependencies and data sources

SinglePointRNA is a Shiny [1, 2] based application that provides a graphical interface to two single-cell analysis R packages, Seurat 4.0 [3] and scCATCH [4].

The *Altered Pathways* module makes use of several MSigDB v7.5.1 [5] collections of gene-sets characterizing metabolic pathways annotated in public databases: Biocarta [6], Pathway Interaction Database (PID) [7], Reactome [8], and WikiPathways [9]. Each of the genesets is named with a prefix indicating its origin.

The following datasets are used in the accompanying materials:

- Human Peripheral Blood Mononuclear Cells, 10X Genomics [10].

- Hematopoietic stem and progenitor Cells from mouse bone marrow (GSE81682) [11].

- 4-day mouse embryoid bodies (GSE130146) [12].
- Endothelial cells from lung samples of mice exposed to intermittent hypoxia or ambient air (GSE145436) [13].

The PBMC dataset is also used in this article for demonstration purposes. The count matrix was pre-processed to remove low-quality cells (<1000 features or <4000 read counts). Then, the counts were normalized using SCTransform [14], regressing the percent of read counts corresponding to mitochondrial and ribosomal RNA.

## Co-clustering conservation

The aim of this measurement is to quantify the changes in clustering assignment as input parameters change. Given two adjacent clustering results, co-clustering conservation is defined as the fraction of cluster mates are maintained for each cell from clustering result *x-1* to clustering result *x*. It is calculated as follows:

$$\mathbf{CC_{i_x}} = \frac{\sum_{j=1}^{J} (\mathbf{CCM_{x-1}}[j,i] = 1 \wedge \mathbf{CCM_x}[j,i] = 1)}{\sum_{j=1}^{J} (\mathbf{CCM_{x-1}}[j,i] = 1)} \qquad \forall j \neq i \tag{1}$$

With $\mathbf{CCM_x}$ and $\mathbf{CCM_{x-1}}$ being co-clustering matrices of size *IxI*. Each of the elements of a co-clustering matrix is calculated by an indicator function as follows:

$$\mathbf{CCM}[j,i] = 1(i,j) = \begin{cases} 1 & \text{if } Cluster_i = Cluster_j \\ 0 & \text{if } Cluster_i \neq Cluster_j \end{cases} \tag{2}$$

With *Cluster_i* and *Cluster_j* being the cluster assignments for cells *i* and *j* in a given run of the algorithm.

## Clustering Uncertainty Score

Using a random sample of seeds for a given set of parameter values to test, the *Clustering Uncertainty Score* or CUS is calculated for each cell in the dataset as follows:

$$CUS(Cell_i) = \frac{1}{J*N} \sum_{j=1}^{J} \sum_{n=1}^{N} \left( 1(n) = \begin{cases} 1 & \text{if } Cl_{i_n} = Cl_{j_n} \\ 0 & \text{if } Cl_{i_n} \neq Cl_{j_n} \end{cases} \right) \qquad \forall j \neq i \tag{3}$$

With $n \in [1, N]$ being each of the iterations of the clustering algorithm, $j \in [1, J]$ being the set of cells co-clustered with cell *i* at any of the iterations, and *Cl_i* and *Cl_j* being the cluster assignments of cells *i* and *j* respectively.

## Pathway enrichment

Molecular pathway enrichment in differentially expressed genes (DEGs) is calculated using Fisher's Exact Test, with the magnitude of the enrichment being measured as an Odds Ratio:

$$OR(Pathway_x) = \frac{DE_{Path}/non - DE_{Path}}{DE_{non-Path}/non - DE_{non-Path}} \tag{4}$$

With genes consistently expressed divided into four categories depending on their differential expression status (*DE* and *non-DE*) and their inclusion in a given molecular pathway (*Path* and *non-Path*). Both raw and FDR-adjusted p-values for the Fisher's Exact Test are returned as well.

## Results

SinglePointRNA provides a simple graphic interface (shown in Fig 1) to perform the basic steps on an analysis of scRNA-seq data: quality assessment, cell filtering, sample integration, cell clustering and differential expression analysis, following the guidelines set by the authors of the Seurat package [3]. Additional processes implemented in this application are described in more detail below.

### Representing variation in clustering results

One of the fundamental goals with this application was to create intuitive ways for novice users to understand the influence of relevant parameters in clustering, namely number of principal components used and clustering resolution, in clustering results. To that end we implemented two ways to visualize change across a range of parameter values: conservation of cells co-clustered together (Fig 2A), and interactive Sankey flowcharts that represent changes in cluster assignment (Fig 2B). The co-clustering conservation plots allow to identify parameter

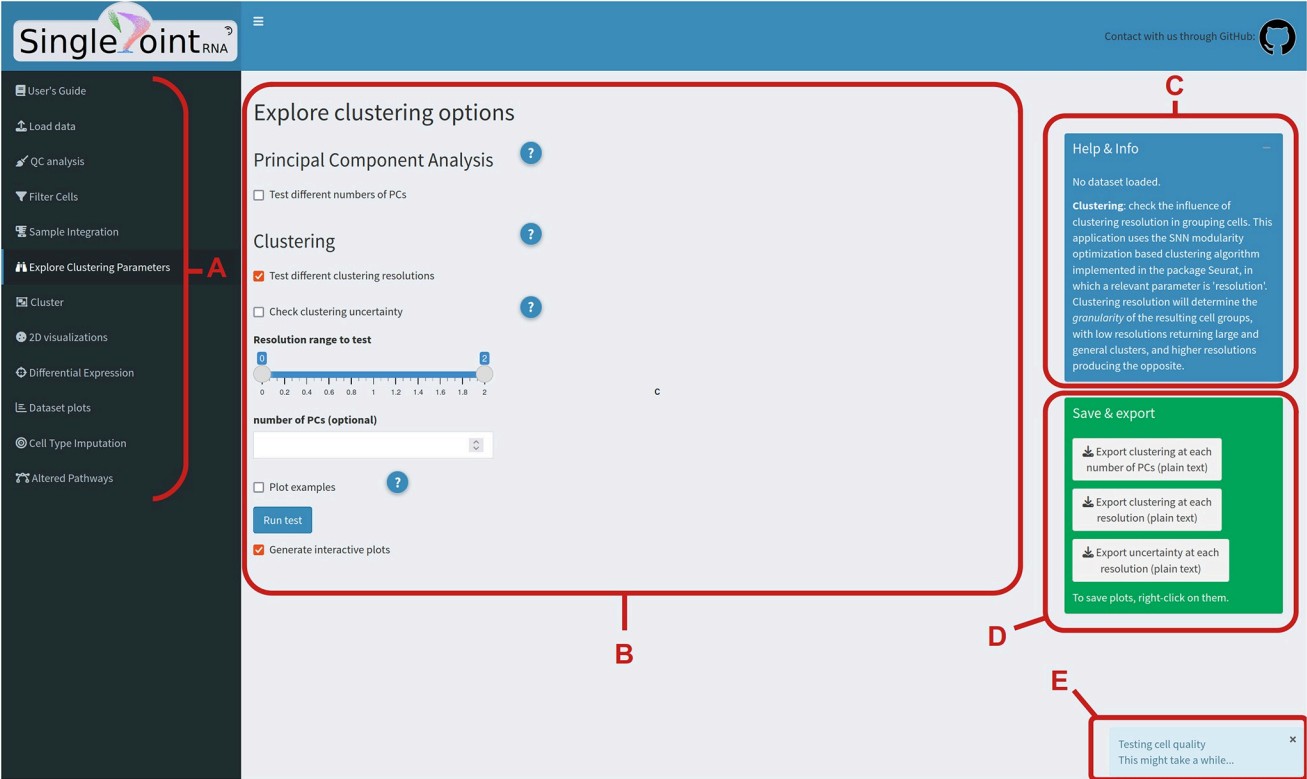

**Fig 1. Interface structure of SinglePointRNA. A**: Each function has its own section accessible through the sidebar. **B**: Main work area. In this section are the widgets for parameter selection. After running each operation, results are displayed below. **C**: Help and information box, contains a brief descriptor of the dataset loaded and displays help messages when a help button is clicked on. **D**: Export box, contains download buttons for result tables and processed datasets. **E**: After an operation is executed, pop-up notifications will appear at the bottom of the screen. Reprinted under a CC BY license, with permission from Fundación Parque Científico de Madrid, original copyright 2022.

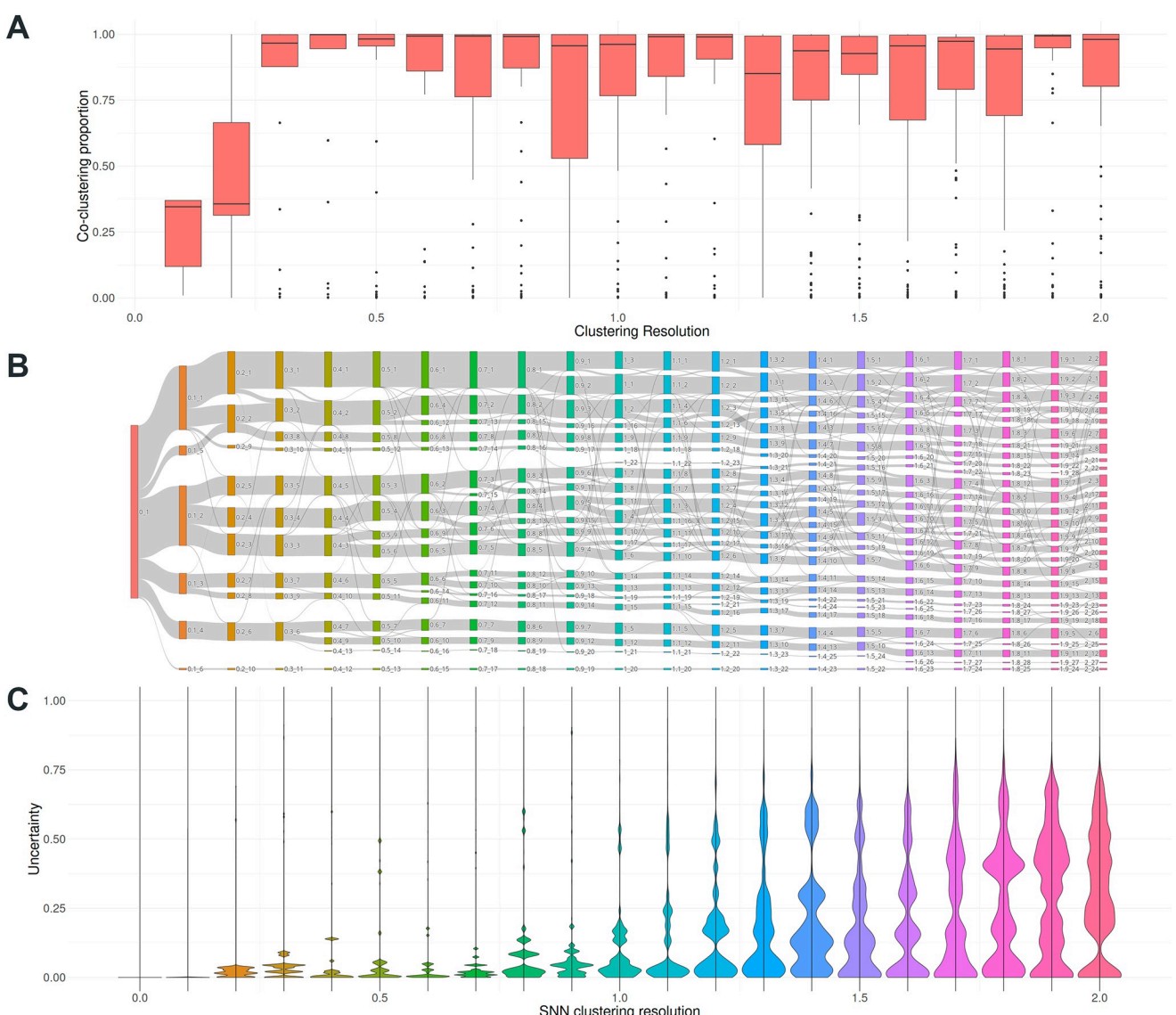

**Fig 2. Example of co-clustering conservation and cluster flow plots.** Example of the graphical outputs for the PBMC dataset, examining changes in clustering for resolutions between 0 and 2. **A:** Co-clustering conservation boxplot. **B:** Clustering flow Sankey diagram. **C:** Violin plots representing the distributions of uncertainty scores of the cells in the dataset at each resolution.

values that result in major changes in cell cluster assignments. Meanwhile, interactive Sankey plots are an ideal medium to represent the merge and division of groups alongside that same range of parameter values.

As an example of how to interpret these two measurements, Fig 2 displays the co-clustering conservation (panel A) and clustering flow plot (panel B) for the PBMC dataset, with clustering resolution values ranging from 0 to 2. Through both representations is clear how the cell population is quickly split between resolutions 0 to 0.2, with some additional adjustments that take place for clustering resolutions up to 0.6. But as can be seen in Fig 2B, when increasing the resolution value further, very few of the rearrangements remain stable. This information, taken together with the Clustering Uncertainty Scores (Fig 2C) facilitates the identification of

clustering parameters that better fit the underlying structure of the data, depending on the level of detail the user is interested on.

## Clustering Uncertainty Scoring

To further aid users with parameter tuning, and taking advantage of the stochastic component of the clustering algorithm implemented in Seurat [15], we have defined the variable *Clustering Uncertainty Score* (CUS), which functions as a measure of similarity between several clustering runs.

Thus, CUS helps users find which clustering resolution values return more robust classifications of the cells in the dataset. Specifically, CUS makes it easier to identify cells that are co-clustered with different sets of neighbors across several iterations of the clustering algorithm, or *ambiguously clustered* for short.

As depicted in Fig 3, the CUS of a cell will depend on two parameters: the size of the conflicting clusters, and how frequently the cell is assigned to each cluster. As it's currently implemented, uncertainty scoring heavily penalizes small groups of cells that are inconsistently

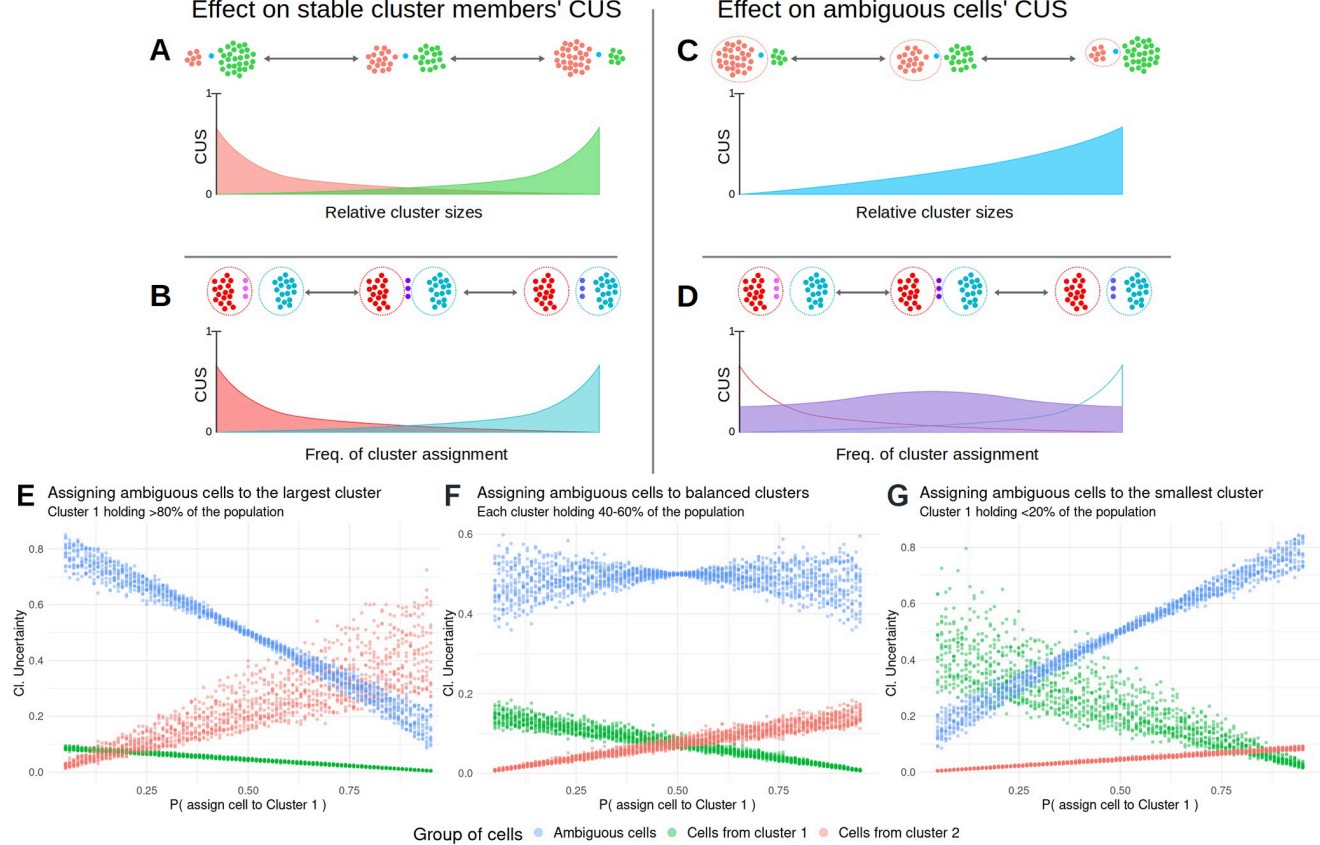

**Fig 3. Clustering Uncertainty Score's behavior.** Considering a population composed of cells classified consistently in one of two clusters, and a cell or small group of cells that are assigned ambiguously. **(A-D)** Schematic representation of CUS changes on the stable cluster members (A-B) and the ambiguous cells (C-D). Panels **A** and **C** display the influence of cluster size: if the considered clusters are of similar sizes, uncertainty scores will be limited to moderate values, while if these clusters are of disparate size the range of CUS expands towards more extreme values. Panels **B** and **D** explore the influence of the frequency with which ambiguous cells are assigned to one cluster or another: ambiguous cells increase their CUS moderately as the assignment frequency gets closer to parity, while on the stably cluster cells, infrequent losses of cluster partners have a higher penalty. **(E-G)** Simulation of CUS scores for a population of 30 ambiguous cells and 220 consistently clustered cells according to the probability of being assigned to the larger cluster (E), the probability of being assigned to one of two balanced clusters (F), and the probability of being assigned to the smaller cluster (G).

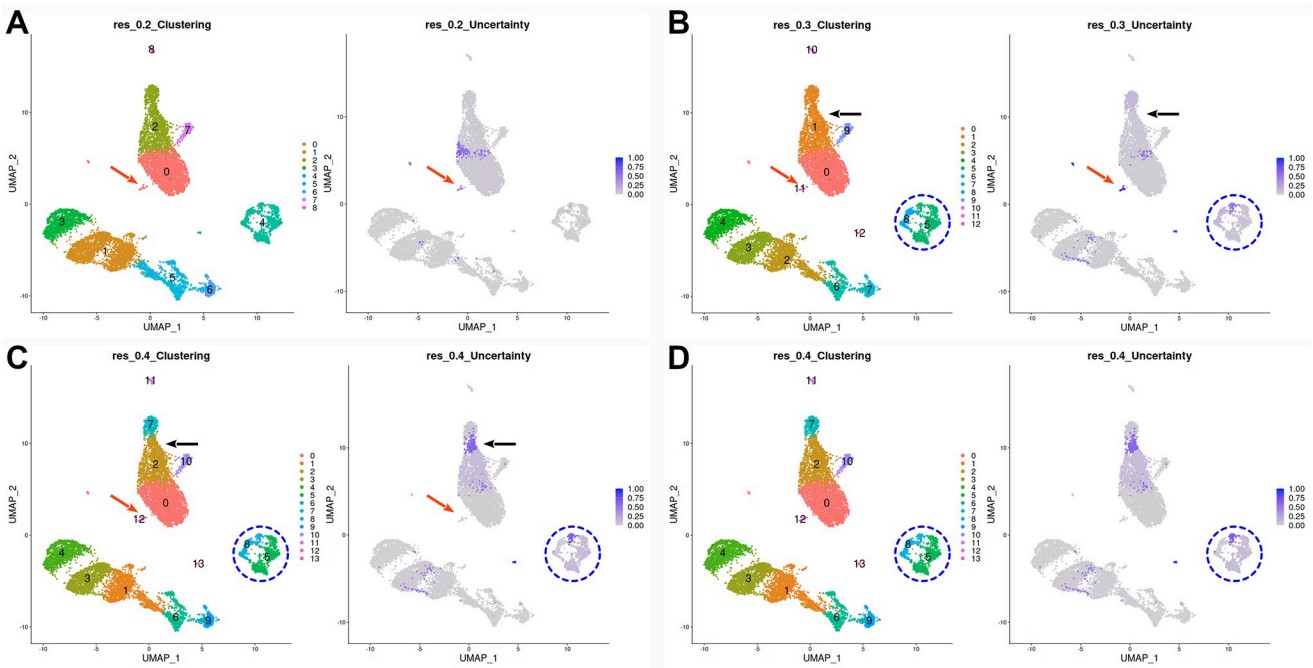

**Fig 4. Example of Clustering Uncertainty Scores in a PBMC dataset.** Clustering results and uncertainty scores for the 10K PBMC dataset at different clustering resolutions. **A**: resolution = 0.2. **B**: resolution = 0.3. **C**: resolution = 0.4. **D**: resolution = 0.5. Red arrows mark the split of a small cluster of cells from cluster A-0. Black arrows highlight a case of and unstable border when cluster B-1 is divided into two other clusters (C-2 and C-7), which boundaries are re-arranged when using a slightly higher clustering resolution (D-3 and D-6). Finally, the blue circle marks a group of cells whose classification into clusters maintains its ambiguity at different resolution values.

excised from a large cluster (Fig 4, red arrows), and moderately penalizes unstable borders between clusters of similar size (Fig 4, black arrows). This dual behaviour has the advantage of being able to display explicitly two major reasons for variation in clustering results at a glance: gradual changes across a group of cells, in which the cluster boundaries will show moderate CUS scores, and inconsistent excisions of a small fraction of a cell population. On a dataset-wide view, lower CUS across the cells will indicate that the clustering parameters chosen produce results that match the underlying structure of the data more closely in an intuitive manner for the user. An additional example applied to parameter tuning is shown in Fig 2C, which shows the distribution of CUS across different values of clustering resolution: for resolutions over 0, uncertainty values show local minima at values of 0.1, 0.4, 0.8, and 1.3, which highlights them as suitable parameter values to classify the cells at different levels of detail, from a broad separation into major cell types to detailed sub-populations within peripheral blood mononuclear cells.

In addition to that, persistent moderate to high values of CUS in a group of cells might also indicate that the genes used to characterize the whole dataset are not enough to properly define stable subgroups (Fig 4, blue circle). In that case, the recommendation would be subsetting these cells and analyze them separately.

## DEGs and biologically meaningful knowledge

Characterizing the changes in gene expression between cell populations (also known as *differential expression* or *DE*) is one of the main goals of studies making use of RNA-seq

technologies, and single cell RNA-seq is no exception. SinglePointRNA implements the default test for DE used in Seurat, based on the Wilcoxon Ranked Sum test. The app includes the possibility to make complex comparisons between groups of cells using one or two of the variables in the cell metadata (for example "*cluster*" and "*treatment*") in three modes (Fig 1A, "Differential Expression":

- 1 VS rest: comparing members of a group to the rest of the cells available in the dataset, which is the conventional formula for DE.

- 1 VS 1: pairwise comparisons between cell groups.

- Conditional DE: when using two grouping variables, it will do pairwise comparisons between each value in the first variable restricted to each value of the second variable. For instance, using *treatment* and *cluster*, the output will include comparisons of every type of treatment for each cluster independently.

Given that DE analysis outputs tend to be arid and devoid of context, sometimes the results can be of limited use by themselves. Thus, we have implemented two functionalities in Single-PointRNA that aim to provide biological insight to DE outputs:

- Cell type imputation using the automated annotation system provided by the R package scCATCH [4] or letting the user define lists of relevant markers of interest (Fig 1A, "Cell Type Imputation").

- DEG enrichment in gene sets defining particular molecular pathways, sourced from MSigDB's collection of Canonical Pathways [5] (Fig 1A, "Altered Pathways").

## Discussion

### A GUI to open scRNA-seq analysis to non-bioinformaticians

The initial motivation of this project was to make the analysis of scRNA-seq data-sets available to researchers with a limited bioinformatic knowledge and thus avoiding the requirement to write code. SinglePointRNA is an interface to the state-of-the-art tools to analyze this type of datasets that substitutes the command line interface for a graphical one, structured in such a way that guides the user throughout the process of analysis. In addition, the application includes a set of tools that aid the user in the selection of appropriate parameter values. Although this is particularly useful for novice users, these functions are not included in widely used packages for scRNA-seq analysis and, as such, could also be useful for bioinformaticians in both trying to determine an optimal set of parameters in an unbiased way, or improving cluster characterization by including uncertainty scores as a latent variable on differential expression analysis. Finally, SinglePointRNA also includes a set of functionalities that try to add biological insight (cell imputation and molecular pathway alteration) to the analysis's results.

### A pedagogical approach to bioinformatic analysis

Extensive research has been done in the fields of pedagogy and mathematical education with regards to the challenges in acquisition of quantitative and computational skills from secondary to tertiary education [16–18], which can entail an additional barrier to the adoption of experimental techniques that require use of high-dimensional data analysis protocols. In addition to giving autonomy to less experienced researchers when in comes to obtaining results from their experiments, a fundamental goal of this project is to assist the users in learning the

conceptual foundations of scRNA-seq analysis. In order to help novice users to get to know the analysis steps and their relevant parameters, all modules of SinglePointRNA include comprehensive help messages within each tab. Beyond that, a general User's Guide, as well as step by step tutorials for different experimental settings are also included. Case datasets and other materials to follow the tutorials have also been added to our GitHub repository. Furthermore, in order to ease getting a better understanding of the basis behind different steps of the process, several general introductions to computer science and machine learning concepts are also included in the app, covering topics such as dimensional reduction, clustering, or optimization. With this content, structured within a pedagogical framework, our objective is that at the end of the day the user is not kept limited to reproducing the steps prescribed in a protocol, but that they manage to internalize the reasoning behind the process and understand the nuances and limitations in interpreting the results obtained. At the same time, we hope that a more gentle and contextually familiar introduction to data analysis theory is able to foster confidence of users in their capability to learn about and apply bioinformatics techniques, with SinglePointRNA being a platform that motivates them to broaden their horizons further.

## Modularity and adaptability

Our choice to build SinglePointRNA as a Shiny app provides great flexibility of use. The software can be run locally from RStudio [19], installed in a shared server and made available to a wide array of users through a web browser, or alternatively, included in a Docker container. The application itself is structured in a modular fashion (Fig 5), trying to keep each phase of the analysis as independent as possible, in order to facilitate modifications of updates on each of the steps without altering the functioning of others. This modularity also makes easier to update the application to include new functionalities that might be of interest for a particular group of users. This would also allow SinglePointRNA to act as a hub in which bioinformaticians could implement their single-cell analysis methods to make them more accessible to collaborators or to the wider research community.

## Semi-blind analysis

In life sciences *blind analysis* is often referred to under the context of clinical trials, and occasionally linked to sample collection broadly, but experimenter effects, as well as publication biases are well documented phenomena [20–22], which should not be understated when it comes to bioinformatics analysis. Moreover, the successes of blind analysis pipelines in other fields [23, 24], both in avoiding discarding results that deviate from available literature and minimizing observer biases, prove that there is much to gain from incorporating blind analysis principles in data collection and analysis into life sciences in general. To that end, the choice was made to adapt SinglePointRNA to follow some basic principles to avoid possible biases while keeping the software accommodating to novice users. First and foremost, by providing methods for parameter tuning for clustering that do not rely in visual inspection of the cluster assignments through tSNE or UMAP plots. In addition to that, most graphical representations as a whole have been restricted to the *Dataset plots* section of the application, with other dataset representations remaining optional. When merging or integrating different datasets, sample-wide metadata can be codified to obscure information that could lead on the user on what results to expect.

## Comparison between SinglePointRNA and other existing tools

The growing prominence of single-cell RNA sequencing (scRNAseq) has spurred the development of various tools dedicated to simplifying data visualization and analysis. Notable among

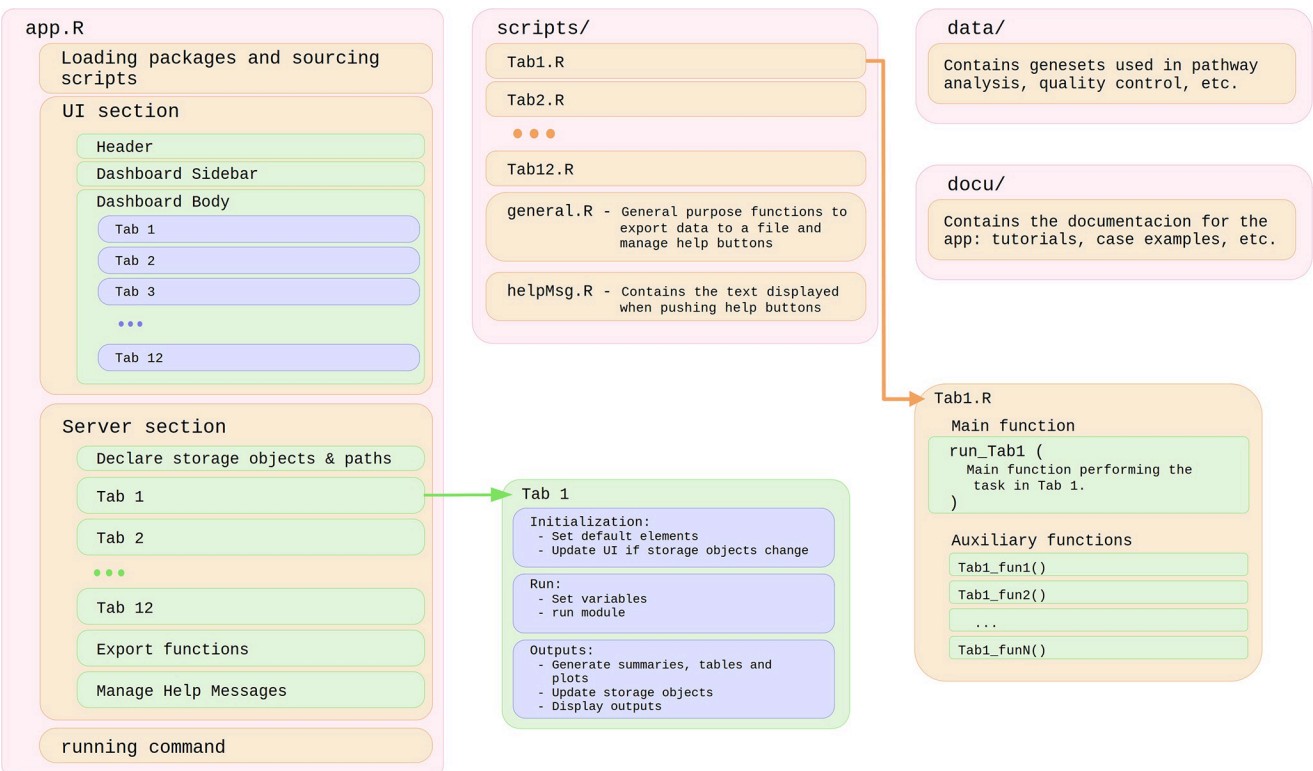

**Fig 5. SinglePointRNA implementation overview.** SinglePointRNA is composed of four main sections. The script "app.R" contains the core of the program. Structured in two blocks, "UI" and "server", as most shiny apps are, "app.R" sets the interface elements of each tab, as well as handles the inputs and outputs of each task. The "scripts/" folder contains all the code needed to perform each task, with one self-contained script per tab. The "data/" folder holds gene set lists and gene naming patterns used for pathway analysis and quality control, stored in plain text files so they can be easily expanded. Finally, the "docu/" folder stores the documentation displayed in the first tab of SinglePointRNA: the user guide, tutorials, etc.

these are ShinyCell [25], scViewer [26], and SCHNAPPs [27]. Here, we discuss the distinctive features of SinglePointRNA in comparison to these tools, emphasizing its contributions to the state of the art in scRNAseq analysis.

ShinyCell, along with counterparts like scViewer, primarily focuses on sharing and visualizing preprocessed single-cell data. These tools serve as valuable complements to Single-PointRNA, as they demand preprocessed data as input and emphasize data visualization. In contrast, SinglePointRNA offers a more comprehensive approach, placing a strong emphasis on scRNAseq data processing, interactive visualization, and functional insights. This broader utility positions SinglePointRNA as a versatile tool for researchers seeking a comprehensive solution.

While SCHNAPPs encompasses essential functions for complete scRNAseq data processing, it lacks certain features that enhance biological interpretability. In comparison, Single-PointRNA introduces functionalities such as functional enrichment analysis of differentially expressed genes and automatic cell type imputation. In addition the incorporation of two novel metrics, co-clustering conservation and clustering uncertainty score, along with interactive Sankey flowcharts, represents a significant advancement. These features aid users in making informed decisions on clustering resolution, a crucial parameter for robust clustering outcomes. These unique characteristics set SinglePointRNA apart, adding depth and biological relevance to scRNAseq analysis.

During the manuscript submission, an additional tool named SEQUIN [28], designed for non-bioinformaticians conducting complete raw scRNAseq analysis and sharing similarities with SinglePointRNA, was introduced. Both tools facilitate parameter selection and functional enrichment analysis of differentially expressed genes through dedicated graphs. However, a crucial distinction arises in their deployment approaches. SinglePointRNA operates primarily as a stand-alone tool, allowing for local installation and operational flexibility. In contrast, SEQUIN leans towards web-based functionality, potentially imposing limitations on the scale of analyses. Additionally, SinglePointRNA's novel metrics offer deeper insights into clustering performance, providing researchers with a comprehensive toolkit for robust analysis.

Thus, SinglePointRNA stands out as a significant advancement in scRNAseq analysis, showcasing comprehensive processing capabilities, unique metrics, and flexible deployment options. Notably, its emphasis on empowering researchers with informed clustering decisions and enriched biological interpretations positions SinglePointRNA as an invaluable asset for the scRNAseq analysis community. The incorporation of novel metrics and a user-centric approach reflects its commitment to providing a robust toolkit for researchers seeking deeper insights into their single-cell RNA sequencing data.

## Conclusion

SinglePointRNA is a Shiny based application providing access to scRNA-seq analysis tools to users without previous experience in programming or computer science in general. In addition to offering a graphic interface, the application includes ample guidance materials, as well as the choice to modify a wide array of parameters, giving users the chance to conduct moderately complex analyses in a transparent and user-friendly environment. Beyond serving as a graphical interface to published tools for scRNA-seq analysis such as Seurat and scCATCH, the application includes several additional functionalities, aiding in parameter selection for cell clustering and connecting differences in gene expression to biological function, useful to a wide range of users from novice to experts.

SinglePointRNA is designed for users and learn along the way as they use the app, both on the particular topic of single cell RNA-seq analysis as well as on related topics on machine learning and computer science in general. Besides its pedagogical interest, this application also facilitates to reduce biases in the analysis procedure by providing methods of parameter selection that are not reliant on direct inspection of the results, which could could be a particularly vulnerable step to visual perception limitations and biases.

Finally, SinglePointRNA allows for cross-platform and flexible implementation in both individual computers as well as shared networks, and its modular structure makes it easy to update and supplement with additional functionalities.

## Author Contributions

**Conceptualization:** Laura Puente-Santamaría.

**Data curation:** Laura Puente-Santamaría.

**Funding acquisition:** Luis del Peso.

**Investigation:** Laura Puente-Santamaría.

**Project administration:** Luis del Peso.

**Software:** Laura Puente-Santamaría.

**Supervision:** Luis del Peso.

**Validation:** Luis del Peso.

**Visualization:** Laura Puente-Santamaría.

**Writing – original draft:** Laura Puente-Santamaría.

**Writing – review & editing:** Laura Puente-Santamaría, Luis del Peso.

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
